# Magnon-polaron control in a surface magnetoacoustic wave resonator

Kevin Künstle [1] ✉, Yannik Kunz [1], Tarek Moussa[1], Katharina Lasinger [1,2], Kei Yamamoto [3], Philipp Pirro [1], John F. Gregg[2], Akashdeep Kamra [1] & Mathias Weiler [1]

Strong coupling between distinct quasiparticles in condensed matter systems gives rise to hybrid states with emergent properties. We demonstrate the hybridization of confined phonons and finite-wavelength magnons, forming a magnon-polaron cavity with tunable coupling strength and spatial confinement controlled by the applied magnetic field direction. Our platform consists of a low-loss, single-crystalline yttrium iron garnet (YIG) film coupled to a zinc oxide (ZnO)-based surface acoustic wave (SAW) resonator. This heterostructure enables exceptionally low magnon-polaron dissipation rates below $\kappa/2\pi < 1.5$ MHz. The observed mode hybridization is well described by a phenomenological model incorporating the spatial profiles of magnon and phonon modes. Furthermore, we report the first observation of Rabi-like oscillations in a coupled SAW-spin wave system, revealing the dynamical formation of magnon-polarons in the time domain. These results establish a platform for engineering hybrid spin-acoustic excitations in extended magnetic systems and enable time-resolved studies of magnon-polaron states.

Recent advances in solid-state fabrication and design techniques have enabled hybrid systems that bear properties and phenomena that are not supported by their constituent subsystems alone[1–4]. When excitations or quasiparticles existing in the two constituent subsystems couple together linearly, a hybrid quasiparticle inheriting properties from both parents is formed[5,6]. This is achieved only when the parent quasiparticles interconvert more rapidly than they are lost to dissipation, and this comparison defines the strong coupling regime[7]. Such quasiparticle engineering has enabled a wide range of phenomena, including the Bose-Einstein condensation of exciton polaritons[8–10], where the excitonic component provides the mutual interactions required for these hybrid quasiparticles, as well as the formation of self-hybridized exciton polaritons[11], due to the efficient photonic confinement via its coupling to the excitonic mode.

The low dissipation rate of the bosonic quasiparticles in ordered magnets, namely magnons, has facilitated the formation of their hybrids with photons (magnon-polaritons)[12–18] and with phonons (magnon-polarons)[19–24]. Even quasiparticles combining all three fundamental excitations have been demonstrated in polaromechanical systems[25]. This has opened doors for quantum manipulation of magnets and controlling magnonic spin transport[26–31].

To observe magnon-polariton formation, the large velocity mismatch between the two excitations necessitates hybridization within a microwave cavity[13]. Magnon-polarons hold an advantage in the close proximity of the phononic and magnonic dispersion curves, implying a maximal wavefunction overlap at a crossing point, which however has to be resolved both in time and space[19]. While strong coupling has been demonstrated for uniform magnon modes[21,32], the associated device architectures are not ideal for magnonic applications. Recent reports using surface acoustic wave (SAW) devices are promising, but remain limited by high spin wave (SW) relaxation rates ($\kappa_m/2\pi \approx 50$ MHz in CoFeB[22]), which preclude time-domain studies and other coherent control schemes. To overcome this limitation, we employ high-quality yttrium iron garnet (YIG), the most

[1]Fachbereich Physik and Landesforschungszentrum OPTIMAS, Rheinland-Pfälzische Technische Universität Kaiserslautern-Landau, Kaiserslautern, Germany. [2]Clarendon Laboratory, Department of Physics, University of Oxford, Oxford, United Kingdom. [3]Advanced Science Research Center, Japan Atomic Energy Agency, Tokai, Japan. ✉e-mail: kuenstle@rptu.de

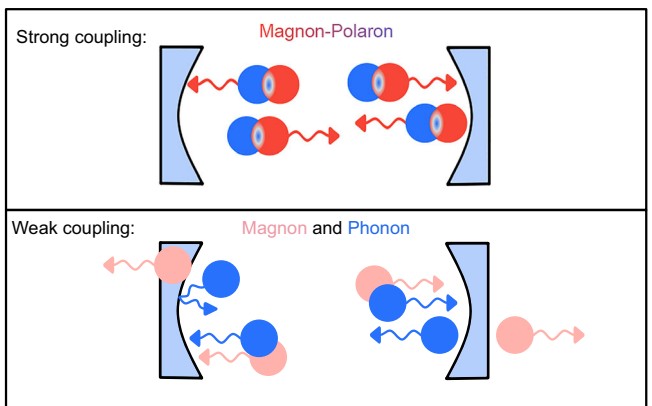

**Fig. 1 | Schematic illustration of quasiparticle confinement in strong and weak coupling regimes.** The acoustic cavity only acts on the phonon (blue) part of the quasiparticle. In the strong coupling regime, the hybrid magnon-polaron is confined by the acoustic cavity even though it does not directly act on the magnonic part (red). In the weak coupling regime, magnons are not confined effectively by the resonator, leading to a weaker magnon wavefunction (lighter red) due to its delocalization over a larger volume. Our system enables switching between the strong and weak coupling regimes by changing the angle between applied magnetic field and SAW $k$-vector direction.

prominent material in magnonics research, owing to its exceptionally low SW damping[33].

We introduce and engineer the means to achieve the desired strong coupling between a magnon and a phonon in an extended magnet-piezoelectric bilayer using only an acoustic cavity. The employed bilayer is large in the planar directions and our device design converts a small fraction of the available space into a cavity. The strong magnon-phonon coupling is expected to enable our fabricated acoustic cavity to also confine the magnon as schematically depicted in Fig. 1. Our magnon-polaron cavity admits a tunability of the coupling strength, experimentally realized via changing the applied magnetic field direction. In this way, we can tune the coupling strength and consequently the quasiparticle nature such that the magnon-polaron is dissociated into its constituent magnon and phonon quasiparticles. As illustrated in Fig. 1, in this weak coupling case, only the phonon mode remains spatially confined in the cavity, while the magnon mode is allowed to escape.

To experimentally realize this concept, we employ a micro-fabricated SAW Fabry-Pérot-type one port resonator defined on an extended bilayer of high quality YIG, a ferromagnetic insulator, and zinc oxide (ZnO), a piezoelectric material. Leveraging the exceptionally low magnetic damping of YIG in combination with a high-quality-factor acoustic resonator enables strong magnon-phonon coupling, manifested as an anticrossing in the dispersion, with matched magnon and phonon loss rates of $\kappa/2\pi < 1.5$ MHz. Supported by theoretical modeling, our system exploits the interplay between SAW phonons and spin wave (SW) magnons, revealing an angle-dependent coupling strength governed by the combined effects of magnetoelastic interactions and the anisotropic spin wave mode profile. Furthermore, time-domain analysis[34] reveals the first observation of Rabi-like oscillations in a strongly coupled SAW-SW system. These oscillations occur on timescales comparable to the transient response of the acoustic resonator. The intrinsically slower propagation of phonons, compared to photons, provides unique temporal resolution and opens new opportunities for dynamically controlling magnon-phonon coupling via tailored, time-resolved driving schemes.

## Results and discussion
### Strong magnon-phonon coupling
The low magnetic damping of YIG is contingent on high-quality monocrystalline films, typically grown on gadolinium gallium garnet

(GGG) substrates[35]. Consequently, conventional integration of magnetic thin films into SAW delay lines on piezoelectric substrates[36,37] is not feasible. To overcome this limitation, we employed a ZnO/YIG bilayer with GGG as the substrate, leveraging the demonstrated efficiency of SAW-mediated SW generation in ZnO/YIG heterostructures[38]. One-port SAW resonators were then fabricated on the ZnO layer, using electron-beam lithography. The piezoelectric ZnO interconverts elastic strain and voltages thereby providing an interface between the generating or detecting electrical circuitry and the SAW mode. The resonator features two opposing, shorted Bragg mirrors and a double-electrode interdigital transducer (IDT) centered between the refractors, which act only on the acoustic mode due to ZnO's piezo-electricity. The IDT and Bragg mirrors were composed of 5 nm Ti and 30 nm Au. The dimensions of the resonator are given in Fig. 2a, and the sample stack including layer thicknesses is shown in Fig. 2c. Our acoustic resonator is defined on an extended bilayer sample (Fig. 2b) and thus occupies a small fraction of the available space to which the hybridized excitations are confined by the resonator. This contrasts sharply with the typical concept and design in magnon-photon systems[39,40], in which the magnons and photons are confined separately by their respective full sample spaces.

Spatially resolved microfocused Brillouin Light Scattering spectroscopy confirms the supported SAW wavelength within the resonator to be $\lambda = 2.09\,\mu m$, agreeing with the defining geometrical restriction (Supplementary Note 1A). To characterize the system eigenmodes, the IDT was connected to a vector network analyzer (VNA), and reflection parameter ($S_{11}$) measurements were performed with the angle $\phi$ between the SAW wave vector $k$ and the external in-plane magnetic field ($B_{ext}$) fixed at $\phi = 0°$ (see Fig. 2a). All measurements discussed here were conducted at a low microwave power of −30 dBm to ensure the system remained within the linear regime. Figure 2d shows the unperturbed resonator's response at $B_{ext} = −5$ mT, where no SAW-SW coupling is observed. The resonator exhibits three distinct acoustic modes within its first stopband.

For further analysis, we focused on the central high quality factor (high-$Q$) SAW mode, which exhibited a resonance frequency of ~1.4 GHz and a quality factor $Q = 1093 \pm 8$, determined from the fit to Eq. (3). From this total quality factor, the SAW loss rate ($\kappa_p$) was determined using the relationship $\kappa_p = \omega_p/Q$, resulting in a value of $\kappa_p/2\pi = (1.28 \pm 0.01)$ MHz, where $\omega_p$ is the angular frequency of the central high-$Q$ mode.

When the external magnetic field $B_{ext}$ is varied, additional field-dependent modes appear alongside the three aforementioned SAW resonator modes. These modes are symmetric with respect to the magnetic field direction, meaning they occur identically at positive and negative field values, as seen in Fig. 2e. In the range between −12 mT and +12 mT the magnetic excitations do not significantly interact with the SAW and are directly excited via the microwave currents in the electrical structures[38]. The dashed lines mark the hybridizing phonon-magnon modes. These arise when the magnon-phonon resonance conditions are fulfilled, leading to multiple anticrossings, highlighted by the solid lines in Fig. 2e. The observation of these avoided crossing features in the reflection data demonstrates that the strong magnon-phonon coupling limit is reached. We identify the specific magnon modes that hybridize with the SAW phonons by comparing the experimentally observed anticrossing positions with theoretically calculated SW dispersion curves for the YIG film (dashed lines in Fig. 2e). We calculate these theoretical dispersions following the approach laid out by Kalinikos and Slavin[41,42], computed for a fixed in-plane $k$-vector equivalent to the SAW wave vector. Our calculation disregards the complex pinning conditions and anisotropies, which we compensate for with a small field shift.

The most pronounced anticrossing occurred at the highest magnetic field, which corresponds to the fundamental ($n = 0$) backward volume spin wave mode. Additionally, we observed clear anticrossings

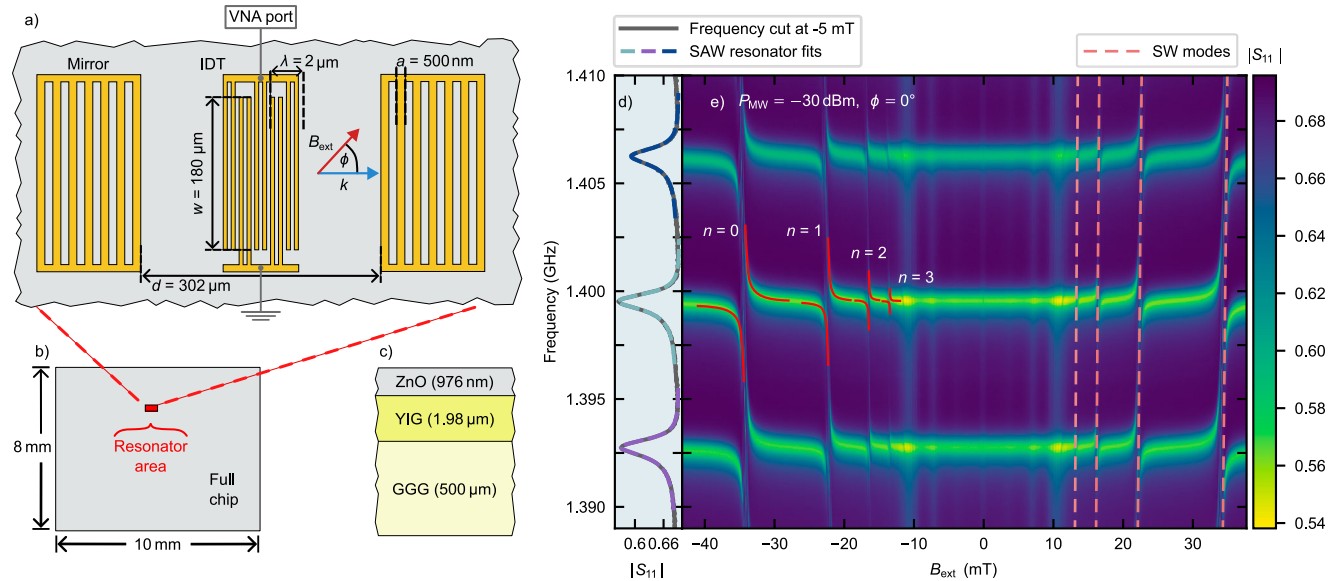

**Fig. 2 | Magnon-polaron formation in acoustic resonator. a** Schematic top view of the fabricated one-port SAW resonator. Mirrors have 100 electrodes each, and the IDT consists of 20 finger pairs. Key dimensions: aperture ($w$), mirror finger spacing/width ($a$), IDT finger spacing/width ($a/2$), resonator length ($d$). The IDT is connected to the port of a VNA. The electrodes forming the resonator are patterned onto an extended multilayer. **b** Schematic top view of the chip, with the resonator region highlighted by a red rectangle. The resonator occupies only a small fraction of the total chip area. **c** Side view schematic of the material stack. **d** $|S_{11}|$ reflection parameter of the uncoupled resonator, showing three high-$Q$ SAW resonances fitted to Eq. (3). **e** $|S_{11}|$ reflection parameter as a function of frequency and $B_{\text{ext}}$. Anticrossing positions align with calculated SW modes (light red dashed lines). Anticrossing of the central high-$Q$ mode is fitted to Eq. (1) for the first four SW modes to obtain coupling strength $g$.

**Table 1 | Observed coupling strength g at $\phi$ = 0° for the fundamental SW excitation and the first three PSSWs**

| SW Modenumber | $n = 0$ | $n = 1$ | $n = 2$ | $n = 3$ |
|---|---|---|---|---|
| $g/2\pi$ (MHz) | 4.81 ± 0.16 | 2.95 ± 0.27 | 1.37 ± 0.31 | 0.58 ± 0.35 |

associated with the first three perpendicular standing spin wave (PSSW) modes ($n$ = 1, 2, 3), thereby demonstrating coupling of phonons also to higher-order spin wave excitations.

To quantify the coupling strength ($g$) between the SAW and these SW modes, we fitted the observed anticrossings in the frequency spectrum to the theoretical expression including damping of the uncoupled modes[39]:

$$\tilde{\omega}_\pm = \frac{\tilde{\omega}_p + \tilde{\omega}_m}{2} \pm \frac{1}{2}\sqrt{(\tilde{\omega}_p - \tilde{\omega}_m)^2 + 4g^2}, \quad (1)$$

where $\tilde{\omega}_{p/m} = \omega_{p/m} - i\kappa_{p/m}$ are the complex angular frequencies of the phonon and magnon, respectively, including their decay rates $\kappa_{p/m}$. The resulting hybridized eigenfrequencies of the coupled system, corresponding to the upper and lower magnon-polaron branches, are denoted by $\tilde{\omega}_\pm$.

Within the narrow frequency window considered, the magnon frequency varies approximately linearly with magnetic field, in agreement with the calculated SW dispersion (dashed light-red lines in Fig. 2e). Accordingly, we model the magnon angular frequency as $\omega_m = \gamma B_{\text{ext}} + c$, with $\gamma = 28 \cdot 2\pi$, GHz T$^{-1}$ the gyromagnetic ratio of YIG and $c$ a constant frequency offset accounting for magnetic anisotropies.

The strongest coupling is observed with the fundamental SW excitation. The extracted coupling strengths for modes $n = 0 - 3$ are summarized in Table 1.

To fully characterize the coupling regime, we determined the magnon loss rate ($\kappa_m$) by analyzing the $n = 0$ SW mode, yielding $\kappa_m/2\pi = (1.24 \pm 0.02)$ MHz. A detailed description of this evaluation,

together with a discussion of additional magnetic features in the anticrossing region, is provided in Supplementary Note 3. Comparing the loss rates with the coupling strength, we confirm that our system operates within the strong coupling regime, with a cooperativity of $C = g^2/(\kappa_m\kappa_p) = 14.5 \pm 1.2$ for the hybridization of the fundamental ($n$ = 0) magnon mode with its corresponding SAW mode.

Our observed coupling strength agrees well with the calculations of our theoretical model (Section "Angle dependence of coupling"), which assumes that the magnon mode is restricted to the magnetic volume within the resonator. This agreement indicates that, in the strong coupling regime, the resonator confines the hybridized magnon-polaron, thereby giving rise to new coupled magnetoelastic boundary conditions[43,44]. Our theoretical model further attributes the observed reduction in the coupling strength for higher SW modes [Table 1] to a decreasing wavefunction overlap with the SAW mode.

### Angle dependence of coupling

We further studied the angular dependence of the coupling strength by varying the angle $\phi$ between the external magnetic field and the wave vector of the excited high-$Q$ SAW mode. We varied the angle in 3° steps and measured the resulting anticrossing behavior. Figure 3a–c shows the anticrossing with the fundamental SW mode $n = 0$ for three exemplary angles 0°, 30°, 60°. The red dashed lines represent fits to Eq. (1), from which the coupling strength ($g$) was determined. The error bars reflect the standard deviations of the fitted parameters as determined by the nonlinear least-squares regression. Evidently, the coupling strength depends on $\phi$.

Before interpreting this $\phi$-dependence, we describe the key features of our experimental design and our theoretical model detailed in the Methods section. In designing our hybrid system, we have chosen the thicknesses of the piezoelectric and the magnetic layers in the micrometer regime because the SAW mode decays on this length scale[45]. In this manner, our design maximizes the overlap between SAW and SW modes, as is also corroborated by our theory. In order to capture the key physics, we modeled the displacement profile ($\mathcal{P}$) of

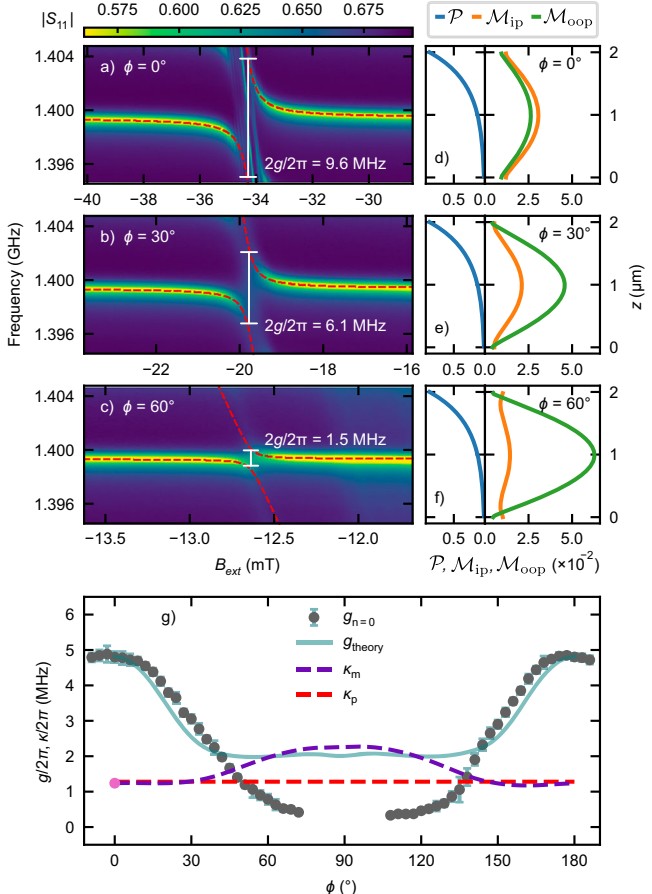

**Fig. 3 | Angle dependence of coupling strength. a–c** $|S_{11}|$ reflection parameter as a function of frequency and $B_{ext}$, showing the anticrosing between the central high-$Q$ SAW mode and fundamental $n = 0$ SW mode at $\phi = 0°, 30°, 60°$. The dashed red lines are fits to Eq. (1). **d–f** Depiction of the modeprofiles $\mathcal{P}$, $\mathcal{M}_{ip}$, $\mathcal{M}_{oop}$ of the phonon and magnon. The magnon mode ellipticity changes substantially with $\phi$. **g** The experimentally measured ($g$) and theoretically calculated ($g_{theory}$) magnon-phonon coupling strengths in dependence of $\phi$ are shown for the spin wave mode $n = 0$. The theory shows excellent agreement around 0° and 180°. The red and purple dashed lines show the loss rates of magnons and phonons, respectively. The single pink data point indicates the directly measured magnon loss rate at 0°.

the standing SAW mode in the resonator as an exponential decay from the surface, accounting for a general three-dimensional acoustic displacement. As demonstrated in refs. [38,46], the excited SAW modes in YIG/ZnO, despite being Rayleigh-type, exhibit shear displacement components which can further be enhanced by the resonator.

The SW mode profiles, which, unlike the SAW profile, depend on $\phi$, were extracted from TetraX[47,48] simulations. We define $\mathcal{M}_{oop}$ as the out-of-plane magnetization component and $\mathcal{M}_{ip}$ as the component orthogonal to the saturation magnetization, which lays in the film plane. The equilibrium magnetization is assumed to be parallel to $B_{ext}$ due to the in-plane anisotropy in YIG being small. The simulated profiles are shown in Fig. 3d–f for the three exemplary angles and detailed simulation parameters are given in Supplementary Note 7. The ellipticity of the SW mode profile changes with increasing angle.

As a key result of our study, Fig. 3g displays the measured (data points) and calculated (solid line) coupling strength as a function of $\phi$ for the fundamental SW excitation ($g_{n=0}$). The angle dependence of the observed higher order couplings is shown in Supplementary Note 5. We observe good quantitative agreement between measurement and calculation. The only fitting parameters in the theoretical analysis are the SAW decay length into the substrate and the ratios between the

different displacement components, since we do not have access to these via experiments or simulation. Nevertheless, the obtained values of these parameters agree well with the general expectations (Supplementary Note 7E).

Our device allows tuning between the strong and weak coupling regime as $\phi$ is varied. The transition occurs when $g$ becomes smaller than the maximum loss rate. The loss rates $\kappa_p$, $\kappa_m$ in our system are shown by red and purple dashed lines in Fig. 3g. A notable aspect of our system is the close proximity of $\kappa_p$ and $\kappa_m$, a characteristic that capitalizes on the low SW damping in YIG.

Our theoretical model takes two key features into account that allow us to understand and adequately reproduce the observed $\phi$-dependence of $g$. First is the mode overlap. Due to the micrometer range thickness of our piezoelectric and magnetic layers, the SW modes significantly change their spatial dependence with $\phi$. Consequently, the wavefunction overlap between the SAW and SW modes is also modified with $\phi$. Second is the SW ellipticity. Since the components of the strain tensor couple differently to the orthogonal SW dynamical magnetizations, a change in SW ellipticity with $\phi$ appears to dominantly influence the $g$ vs. $\phi$ curve. Employing our theoretical model, the observed $\phi$-dependence further shows that the shear strain component dominates the SAW mode supported by the resonator. As $\phi$ increases, the experimentally observed coupling strength (Fig. 3g) decreases more strongly than predicted by our theoretical model, which assumes the phonon and magnon to be confined to the area of the resonator. This additional reduction of coupling strength is in accordance with an increased uncoupled magnon mode volume due to magnon leakage out of the acoustic resonator as depicted schematically in Fig. 1 and discussed in more detail in Supplementary Note 7. These two features that underlie our observations have not been addressed previously and further suggest future avenues for progress, especially by examining the boundary conditions and the hybridization effect.

## Coupling dynamics in time domain

The SAW resonator's standing wave formation and the magnon-polaron hybridization occur on comparable timescales. Thus, the ringing of the cavity phonons and the Rabi-like oscillations of the magnon-polaron bear a similar frequency. While Rabi-like oscillations have been reported in cavity magnon-polaritons[15,49], our hybridized magnon-phonon system thus uniquely allows for the simultaneous observation of these Rabi-like oscillations and the transient evolution of the resonator response.

To access the system's time-domain behavior, we performed an inverse Fourier transform on the complex frequency-domain $S_{11}$ data, obtained from the 0° measurement presented in Fig. 2e, for each applied magnetic field value $B_{ext}$. The resulting time-domain representation, $|\tilde{S}_{11}|$, is shown in Fig. 4a and is equivalent to a conventional time-domain reflectometry signal[34]. In field regions where SAW-SW coupling is negligible, we observe distinct passes of the excited SAW through the resonator, manifesting as periodically spaced horizontally aligned lines that decay exponentially in amplitude. This represents the transient response of the SAW resonator. A corresponding cut along the vertical blue dashed line in Fig. 4a is shown in Fig. 4c (blue line). The periodicity and amplitude decay of these passes allow us to extract key SAW propagation parameters, such as the decay length $l_p = 757\,\mu m$ and group velocity $v_{gSAW} = 2382\,m\,s^{-1}$, as detailed in Supplementary Note 1D.

Conversely, in field regions where coupling occurs (see the close-up view in Fig. 4b), we observe Rabi-like oscillations, an expected signature of a strongly coupled system. An exemplary linecut of the coupled behavior is shown by the orange line in Fig. 4c), revealing a clear change in oscillation period compared to the uncoupled case. Here, the formation of the magnon-polaron becomes apparent within the transient response of the resonator.

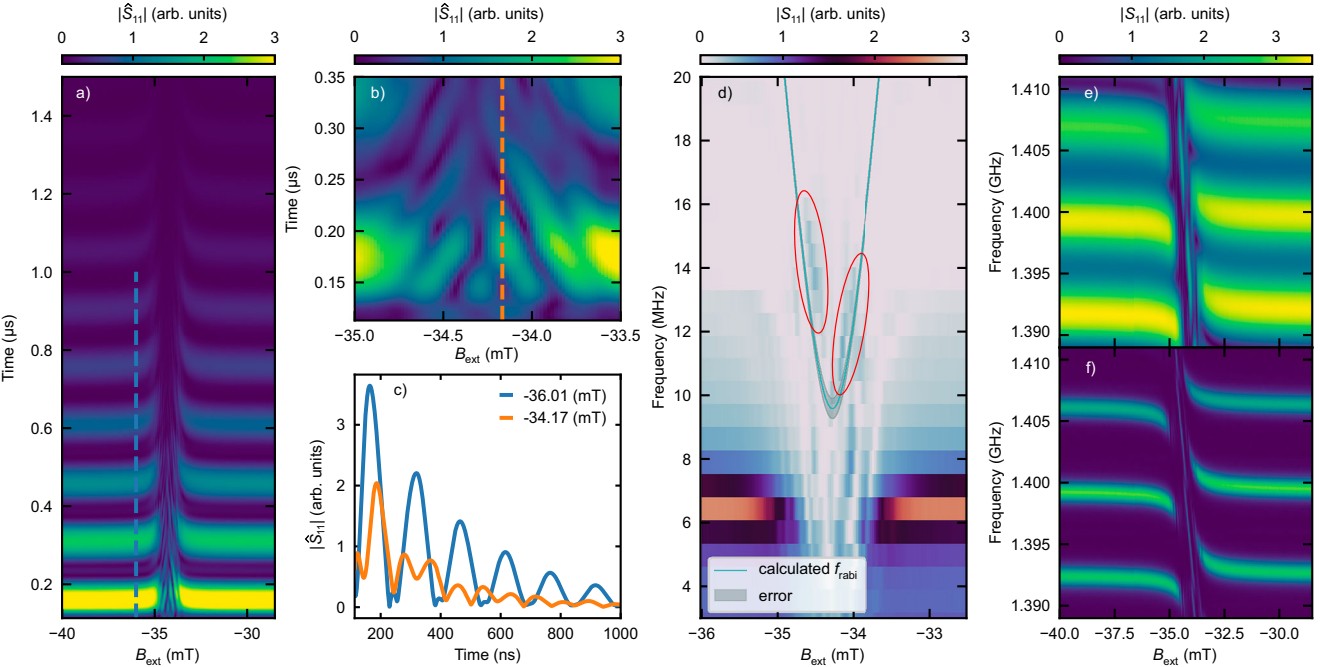

**Fig. 4 | Time dependence of coupling and Rabi-like oscillations. a** Time-domain representation of the $|\hat{S}_{11}|$ reflection parameter as a function of time and $B_{ext}$. **b** Close-up view of Rabi-like oscillations. The dashed orange line indicates the on-resonant time trace used for comparison with the off-resonant case in (**c**). **c** Comparison of on-resonant (orange dashed line from (**b**)) and off-resonant (blue dashed line from (**a**)) time traces, illustrating the change in oscillation frequency. **d** Frequency-domain representation of $|S_{11}|$, obtained via Fourier transform of a

smaller region of interest in time-domain data. The broad horizontal line indicates the SAW resonator frequency. The plotted light blue line represents the calculated Rabi frequency and its numerical error is displayed in light gray. Red ellipses highlight the Rabi frequency signatures in the back-transformed $|S_{11}|$ parameter. **e, f** Time-gated $|S_{11}|$ reflection parameter showing different numbers of SAW passes: **e** 3 passes, **f** 11 passes. A clear dependence of the quality factor on the number of gated passes is observed.

Consistent with the Rabi theory, we observe a change in the oscillation frequency when the energy gap between the two systems is detuned. In our system, this detuning is controlled by the applied external magnetic field strength, which shifts the spin wave dispersion. The theoretically predicted steady-state Rabi frequency $f_{rabi}$ can be evaluated using information solely from the frequency domain analysis presented in Section "Strong magnon-phonon coupling", specifically the detuning $\delta$ and the coupling strength $g$, according to[50]:

$$f_{rabi} = \sqrt{4(g/2\pi)^2 + \delta}, \quad \delta = \frac{\gamma}{2\pi}|B_{ac} - B_{ext}|. \quad (2)$$

Here, $B_{ac}$ denotes the magnetic field at which the anticrossing is centered. The linear relationship assumed for the detuning with the difference between $B_{ext}$ and $B_{ac}$ is justified by the linear change of the SW frequency with magnetic field in this small frequency range, as explained in Section "Strong magnon-phonon coupling". The calculated $f_{rabi}$ is displayed by the blue line in Fig. 4d.

To compare the Rabi frequency from this simplified steady-state model, derived from frequency domain data, with time-domain observations, we performed a Fourier transform to back-transform a smaller region of interest of the time-domain dataset to the frequency domain. The resulting color-coded graph is shown in Fig. 4d. In regions where coupling is absent, the frequency of the SAW ringing is visible as a broad horizontal line at 6.5 MHz. This frequency is in accordance with the expectations from the SAW group velocity and the resonator dimensions. In regions where coupling is present, we identify frequency components in accordance with the Rabi frequency, highlighted by two red ellipses.

Comparing the calculated $f_{rabi}$ from the frequency domain with these highlighted frequency components, we find reasonable agreement with this simplified analysis. The discrepancies can be attributed to several factors. Firstly, the polaron formation occurs concurrently

with the establishment of the SAW mode within the resonator, leading to mutual influence and affecting the emergent polaron. Secondly, additional spurious magnetic signals within the anticrossing region influence the time-domain response of the resonator. The most prominent of these is the direct SW excitation (Supplementary Note 3) at the anticrossing center. Consequently, in the anticrossing center, the theoretically predicted hat-like shape (Supplementary Note 6) is interrupted by a minimum in the $|\hat{S}_{11}|$ signal right at $B_{ac}$, caused by this direct SW excitation. The magnetic-field dependent contrast at lower frequencies in Fig. 4d is attributed to these spurious uncoupled magnon modes.

Finally, we demonstrate the ability to artificially manipulate the cooperativity of the system. This is achieved by applying a window function to a selected number of SAW passes within the resonator and subsequently performing a Fourier transformation to obtain the frequency-domain data. Figure 4e, f shows the frequency-domain signals for 3 and 11 gated SAW passes, respectively, with electromagnetic crosstalk removed (observed on different timescales). A clear broadening of the frequency linewidth of the SAW is observed, indicating that the quality factor decreases with a reduced number of gated passes, which in turn decreases the cooperativity. This illustrates how influencing the transient resonator response can artificially alter the resonator loss rate resulting in a change of cooperativity.

This analysis demonstrates the efficacy of our time-domain method for investigating coupling dynamics, employing readily applicable mathematical techniques detailed in the Methods section, and confirms a comprehensive understanding of the SAW resonator's behavior.

## Discussion

We have demonstrated the hybridization of a magnon-polaron in a virtually infinite magnetic medium, marking a key advance beyond the conventional regime of propagating-mode coupling. Central to our

approach is the use of an acoustic cavity that confines the hybrid excitation when the constituent magnon and phonon modes are strongly coupled, similar to an analogous phenomenon in light-matter interaction[11].

By accounting for both wavefunction overlap and SW ellipticity, our theoretical framework captures the angle-dependent coupling strength observed in the experiments.

The use of a single-crystalline YIG film is essential for attaining comparable loss rates for SAWs and SWs, both below 1.5 MHz−a notable milestone not previously reached in SAW-SW coupling.

In addition, the low phononic group velocity in our system offers a unique opportunity to resolve the coupling dynamics in the time domain. The observation of Rabi-like oscillations provides direct evidence of strong magnon-phonon interaction shaping the temporal response of the SAW resonator. This capability opens new avenues for time-resolved studies and coherent control of hybrid magnon-phonon states through tailored dynamical driving schemes. Moreover, by implementing dissipation engineering strategies[18,25], it may become possible to further suppress magnon-polaron losses and extend the coupling to additional excitations in solid-state platforms, thereby broadening the potential of this on-chip approach.

## Methods

### Device fabrication
The device fabrication began with the deposition of piezoelectric ZnO onto the [111]-grown YIG/GGG substrate using radio frequency magnetron sputtering. Subsequently, a PMMA polymer layer was spin-coated onto the ZnO, and the designed microstructure patterns for the SAW resonator and IDTs were defined in the PMMA using electron beam lithography. Following resist development, the metallic microstructures (5 nm Ti, 30 nm Au) were created by electron beam evaporation. Finally, the PMMA resist mask was removed via a liftoff process. A more detailed description of the device fabrication is provided in Supplementary Note 8.

### Electrical characterization of the SAW resonator
Surface acoustic wave excitation and propagation were characterized using a vector network analyzer (VNA) by measuring the complex reflection scattering parameter $S_{11}$ of the SAW resonator, which is connected via bonding wires to the electric circuitry. In these measurements, the VNA generates an electromagnetic signal at a defined frequency that interacts with the device, and the reflected signal's amplitude and phase are measured from which $S_{11}$ is determined. The quality factor of the supported standing SAWs in the resonator can be evaluated via[51]:

$$S_{11}(f) = \frac{(Q_e - Q_i)/Q_e + 2iQ_i(f - f_0)/f}{(Q_e + Q_i)/Q_e + 2iQ_i(f - f_0)/f}. \quad (3)$$

In this equation, $f$ represents the excitation frequency, $f_0$ the resonance frequency, $Q_i$ the internal quality factor, and $Q_e$ the external quality factor. The total quality factor is given by $1/Q = 1/Q_e + 1/Q_i$. Further properties which can be extracted from the VNA measurement are detailed in Supplementary Note 1B, D.

### Time domain transformation
To analyze the temporal response of the SAW resonators, the measured frequency domain $S_{11}$ data was converted to the time domain ($\hat{S}_{11}$)[34]. This transformation involved applying a Hamming window to the frequency data, followed by zero-padding and an inverse discrete Fourier transform. Time-gating, using a Hamming window, allowed for the isolation of specific time intervals. This was crucial for suppressing early-time electrical crosstalk and for selectively examining individual SAW reflections within the resonator cavity through back-transformation to the frequency domain. The effects of the initial Hamming window were subsequently removed via window de-embedding.

### Theoretical model
The magnon-phonon coupling strength was theoretically determined by considering the overlap between SW and SAW mode profiles. The SW mode profile is accessed using the FEM micromagnetic modeling tool TetraX[47,48], from which the components $\mathcal{M}_{oop}$ and $\mathcal{M}_{ip}$ are extracted. Hereby $\mathcal{M}_{oop}$ refers to the out-of-plane magnetization component and $\mathcal{M}_{ip}$ denotes the magnetization component in-plane orthogonal to the equilibrium magnetization. To account for the standing wave nature of the participating modes, the effective $\mathcal{M}_{ip,oop}$ was obtained by averaging the forward and backward propagating modes.

The mode profile of the standing SAW supported by the resonator is modeled as an exponential decay function [$f(z)$] encompassing all displacement components ($u_{x,y,z}$). Even though this assumption is inconsistent with free-stress boundary conditions, it provides convenient phenomenological parameters in the relative amplitude of the displacement components $t_x$, $t_y$ and $t_z$ with $t_x^2 + t_y^2 + t_z^2 = 1$, to explore the polarization dependence of magnon-phonon coupling.

Following the quantization of phonon and magnon modes, detailed in Supplementary Note 7, and generalizing the approach presented in ref. 52, we obtain the quantum Hamiltonian for magnon-phonon coupling in a cubic [100] lattice after applying the rotating wave approximation, employing ladder operators $\tilde{a}$, $\tilde{a}^\dagger$ and $\tilde{c}$, $\tilde{c}^\dagger$:

$$\tilde{H}_{mec} = -\hbar g_1\left(\tilde{a}\tilde{c}^\dagger + \tilde{a}^\dagger\tilde{c}\right) - \hbar i g_2\left(\tilde{a}\tilde{c}^\dagger - \tilde{a}^\dagger\tilde{c}\right). \quad (4)$$

The ensuing gap in the anticrossing is now given by $2|g| = 2\sqrt{g_1^2 + g_2^2}$ with $g_1$ and $g_2$:

$$g_1 = \frac{k_p}{2}\sqrt{\frac{\gamma}{\rho\omega_p M_s}}\left[b_1 I_{x,ip}(\phi)\sin(2\phi) - b_2 I_{y,ip}(\phi)\cos(2\phi)\right], \quad (5)$$

$$g_2 = \frac{k_p}{2}\sqrt{\frac{\gamma}{\rho\omega_p M_s}}b_2 I_{z,oop}(\phi)\cos(\phi). \quad (6)$$

Here, $k_p$ and $\omega_p$ denote the phonon wavenumber and angular frequency, $\gamma$ is the gyromagnetic ratio of YIG, $\rho$ is the average mass density of the material system in which the phonon is confined, $b_1$ and $b_2$ are the magnetoelastic coupling coefficients of YIG, $M_s$ is the saturation magnetization of YIG and $\phi$ is the angle between SAW $k$-vector and external magnetic field. The overlap integrals $I_{x,ip}$, $I_{y,ip}$ and $I_{z,oop}$ are given by:

$$I_{x,ip}(\phi) = \int_{FM} dz\, t_x f(z)\mathcal{M}_{ip}(z), \quad (7)$$

$$I_{y,ip}(\phi) = \int_{FM} dz\, t_y f(z)\mathcal{M}_{ip}(z), \quad (8)$$

$$I_{z,ip}(\phi) = \int_{FM} dz\, t_z f(z)\mathcal{M}_{oop}(z). \quad (9)$$

A more detailed derivation and description of the theoretical model can be found in Supplementary Note 7.

## Data availability
The data supporting this study are publicly available on Zenodo at https://doi.org/10.5281/zenodo.17365743[53].

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

## Acknowledgements
This work was supported by the European Research Council (ERC) under the European Union's Horizon Europe research and innovation programme (Consolidator Grant "MAWiCS", Grant Agreement No. 101044526), the Deutsche Forschungsgemeinschaft (DFG, German Research Foundation) within the Transregional Collaborative Research Center TRR 173/3-268565370 Spin+X (Project A13, B01, B13) and JSPS KAKENHI Grant No. 21K13886.

## Author contributions
M.W. conceived the project. K.K, K.L. and J-F.G. fabricated the sample. K.K. and Y.K. performed the measurements. T.M., K.K., and A.K. developed the theoretical description of magnon-phonon coupling in discussion with K.Y., P.P., and M.W. The manuscript was written by K.K. with input from all coauthors.

## Funding

## Competing interests
The authors declare no competing interests.
