## [Peer Review File · Nature Communications]

Magnon-polaron control in a surface magnetoacoustic wave resonator

Corresponding Author: Mr Kevin Künstle

Version 0:

Reviewer comments:

Reviewer #1

(Remarks to the Author)

This manuscript reports the observation of strong coupling between phonons and magnons using a microfabricated surface acoustic wave (SAW) Fabry-Perot resonator constructed on a high-quality yttrium iron garnet (YIG)/zinc oxide (ZnO) heterostructure. The SAW resonator exhibits exceptionally low dissipation rates ($\kappa/2\pi < 1.5$ MHz), enabling the formation of hybrid magnon-phonon polaritons. The coupling strength between the SAW and spin wave (SW) is tunable by varying the magnetic field orientation relative to the SAW propagation direction. Through Fourier analysis of the reflection spectra, the authors observed Rabi-like oscillations in the time domain, which characterize the coupling dynamics between SAWs and SWs.

This work has made significant progress in achieving strong coupling between SAWs and SWs, though some explanations in the text remain unclear and require further elaboration.

1. A high-Q SAW mode is crucial for the experiment. How were the ZnO/YIG bilayer and geometric parameters optimized to achieve such high Q-factors? Could the authors provide further explanation on this part?
2. In Figure 2e, the $n=0$ region appears to exhibit multiple SW modes. If these modes indeed exist, how can the respective coupling strengths between these SW modes and the SAW modes be determined?
3. The deviation of g between the theory and experiments for ϕ at around $\pi/2$ (Fig. 3) seems significant. How were the experimental error bars obtained in this section? Could the discrepancy between the theoretical and experimental coupling strengths be further analyzed using, e.g., finite element simulations?
4. The study in Commun. Phys. 5(1), 115 (2022) reported coherent oscillations between magnons and phonons, while that in Phys. Rev. Lett. 130, 193603 (2023) reported swap oscillations between magnons and superconducting qubits. Did the Rabi-like oscillations observed in this work differ from the previous results? Furthermore, can consistent information—such as coupling strengths and magnon decay rates—be extracted in the Rabi-like oscillations signal?
5. The coupling between phonons and magnon polaritons in cavity magnomechanics systems has extensively been studied in, e.g. PRX 11, 031053 (2021) and Nat. Commun. 16, 5652 (2025). Notably, the experiment in the Nat. Commun. 16, 5652 (2025) achieved strong coupling between magnon polaritons and phonons by engineering polariton dissipation, despite the intrinsically weak magnon-phonon coupling strength. Could this dissipation-engineering approach be combined with this work to further enhance the cooperativity between SAWs and SWs?
6. Could you please provide further clarification on how the number of gated passes affects resonator loss? It includes how to experimentally control the number of gated passes.

Reviewer #2

(Remarks to the Author)

This work presents the hybridization of magnons in single-crystalline yttrium iron garnet (YIG) films with phonons confined in a zinc oxide (ZnO)-based surface acoustic wave (SAW) resonator. The study demonstrates tunable coupling strength via

magnetic field direction, achieving strong magnon-phonon coupling through high-Q SAW modes in the GHz regime with low damping rates (~few MHz). A phenomenological model is introduced to describe this interaction, incorporating spatial mode profiles. Rabi-like oscillations are observed in the time domain, providing direct evidence of strong magnon-phonon interaction shaping the temporal response of the SAW resonator. This work is establishing a promising platform for hybrid spin-acoustic systems in extended magnetic materials.

However, several areas require clarification and improvement:

1. Confusing Link Between Damping Mechanisms (Line 060):

The text suggests that the limitation of SAW devices stems from large phonon damping, yet proposes low magnon damping in YIG as a solution. This logical jump is unclear—low magnon damping does not inherently mitigate phonon losses. Please clarify how YIG's low damping addresses the phonon-related limitations.

2a. Unclear Concept of Magnon Confinement (Fig. 1 and Caption):

The claim that magnons are not confined in the weak coupling regime is problematic. Since magnons are excitations within the YIG film, their spatial confinement should not depend on the coupling strength. What does it mean physically for the magnon wavefunction to “escape” the cavity? Please provide experimental or theoretical justification for this assertion.

2b. Ambiguity in “Hybridization-Driven Confinement” (Line 89 & Ref. 31):

The idea that hybridization is necessary for magnon confinement is not clearly substantiated. Drawing a parallel with exciton-polaritons (Ref. 31) is insufficient without rigorous support. Please explain the physical mechanism by which coupling enhances confinement and specify the physical parameter that defines such a confinement.

2c. Ambiguous Discussion of Confinement (Line 212):

The statement that hybridization induces confinement is vague. Since magnons remain spatially within YIG, it is unclear what additional “confinement” is being claimed. Moreover, later discussions (e.g., Line 299) suggest the variation in coupling strength arises from spin wave ellipticity, not spatial effects. The link between ellipticity and confinement needs clarification. I would like to mention that angular-dependent coupling strength was previously used for controlling the magnon-photon coupling, and it governs by the polarization of the photon mode instead of mode confinement [see, IEEE Transactions on Magnetics, 52, 1000107 (2016).]

3. Model Limitations and Suggested Revision (Eq. 1 and Fig. 3):

Equation 1 models the strong coupling regime but fails in the weak coupling region (e.g., near 90° field angle in Fig. 3). To more accurately fit the data, the model should include damping effects, as shown in Eq. (2) of J. Appl. Phys. 129, 201101 (2021). Revising Eq. 1 accordingly would strengthen the analysis.

Recommendation:

The manuscript presents promising and timely results. However, conceptual inconsistencies—particularly surrounding magnon confinement and coupling mechanisms—must be addressed. Revision of the theoretical model and clarification of key physical interpretations are necessary before the paper can be considered for publication.

Version 1:

Reviewer comments:

Reviewer #1

(Remarks to the Author)

The authors revised the manuscript following my comments and suggestions. I recommend this revised manuscript for publication in Nature Communications.

Reviewer #2

(Remarks to the Author)

In my previous report, I raised three issues for the authors' attention, and I appreciate the substantial efforts they have made to address my comments:

1. Confusing Link Between Damping Mechanisms

This issue has been satisfactorily addressed in Line 063. As an optional suggestion, the authors may consider explicitly mentioning the low-damping material (CoFeB) used in Ref. 22 to further clarify the discussion.

2. Ambiguity in “Hybridization-Driven Confinement” and Confusing Discussions

Substantial revisions have been made in both the main text and the Supplementary Material. The presented arguments are now more plausible and self-consistent. I particularly appreciate the clear comparison between cavity magnon–photon polaritons and cavity phonon–phonon polaritons (Lines 123–127). (It would be nice if in future the authors may further improve their device to enable direct spatially resolved measurements of the confinement effect.)

3. Model Limitations and Suggested Revision (Eq. 1 and Fig. 3)

I am pleased to see that the equation I suggested was helpful for the authors to significantly reduce fitting errors in the weak-coupling regime.

Overall, I concur with the authors that their device has the potential to open new avenues for time-resolved studies and coherent control of hybrid magnon–phonon states through tailored dynamical driving schemes. I recommend this work for publication in Nature Communications.

Reply to the Referees: Magnon-polaron control in a surface magnetoacoustic wave resonator

Kevin Künstle,^{1, a)} Yannik Kunz,¹ Tarek Moussa,¹ Katharina Lasinger,^{1,2} Kei Yamamoto,³ Philipp Pirro,¹ John F. Gregg,² Akashdeep Kamra,¹ and Mathias Weiler¹

¹⁾*Fachbereich Physik and Landesforschungszentrum OPTIMAS,
Rheinland-Pfälzische Technische Universität Kaiserslautern-Landau,
67663 Kaiserslautern, Germany*

²⁾*Clarendon Laboratory, Department of Physics, University of Oxford, Parks Road,
Oxford, OX1 3PU, United Kingdom*

³⁾*Advanced Science Research Center, Japan Atomic Energy Agency, Tokai,
319-1195, Japan*

(Dated: 19 September 2025)

^{a)}Electronic mail: kuenstle@rptu.de

Dear Referees,

We thank you for the valuable comments and suggestions. In our reply and revised manuscript, we have taken all suggestions into account. Below please find our point-to-point replies in red.

Sincerely yours,

Kevin Künstle on behalf of all authors.

Reviewer #1 (Remarks to the Author):

This manuscript reports the observation of strong coupling between phonons and magnons using a microfabricated surface acoustic wave (SAW) Fabry-Perot resonator constructed on a high-quality yttrium iron garnet (YIG)/zinc oxide (ZnO) heterostructure. The SAW resonator exhibits exceptionally low dissipation rates ($\kappa/2\pi < 1.5$ MHz), enabling the formation of hybrid magnon-phonon polaritons. The coupling strength between the SAW and spin wave (SW) is tunable by varying the magnetic field orientation relative to the SAW propagation direction. Through Fourier analysis of the reflection spectra, the authors observed Rabi-like oscillations in the time domain, which characterize the coupling dynamics between SAWs and SWs.

This work has made significant progress in achieving strong coupling between SAWs and SWs, though some explanations in the text remain unclear and require further elaboration.

We are grateful to the reviewer for their positive evaluation of our work and for the constructive comments. Below, we provide point-by-point responses and describe the corresponding revisions made to the manuscript.

1) A high-Q SAW mode is crucial for the experiment. How were the ZnO/YIG bilayer and geometric parameters optimized to achieve such high Q-factors? Could the authors provide further explanation on this part?

To estimate the achievable quality factor prior to device fabrication, we relied on analytical expressions provided in Supplementary Note 1C (Equations 4–6), which incorporate both material parameters and geometric constraints. When it comes to geometric constraints one has to account for practical limitations of the electron beam lithography process, such as stitching errors and writing field sizes. As an illustrative example: assuming $N_g = 100$, $f = 1.4$ GHz, $r_s = 2\%$, $a = 500$ nm, $d = 300$ μm , $v_{\text{gSAW}} = 2400$ m s^{-1} and $l_{\text{SAW}} = 600$ μm , we estimate an internal Q factor of $Q_i = 1025$ from Supplementary Note 1C (Equations 4–6). This provides a reasonable first estimate for the expected internal quality factor of the resonator, taking the main design and material parameters into account. The quality factors of our fabricated resonators are in good agreement with these estimates, which thus proved sufficient for resonator optimization. The external quality factor is much larger than the internal quality factor in our design and does not limit the achievable total quality factor.

To further highlight the relevance of geometrical constraints when designing high-Q magnetoacoustic resonators, we now explicitly discuss a design parameter that was not previously highlighted in the manuscript: the ratio of ZnO film thickness to acoustic wavelength, h_{ZnO}/λ . This parameter directly impacts the electromechanical coupling strength of the SAW. To address this, we have revised the first paragraph of Supplementary Note 1 to include the following sentence and an additional reference ([1]):

“Building on Reference [1] and our previous work on the YIG/ZnO heterostructure [2], we selected a ZnO thickness to wavelength ratio of $h_{\text{ZnO}}/\lambda \approx 0.5$ to ensure sufficient electromechanical coupling.”

2) In Figure 2e, the $n=0$ region appears to exhibit multiple SW modes. If these modes indeed exist, how can the respective coupling strengths between these SW modes and the SAW modes be determined?

A zoomed-in view of the region in question is provided in Supplementary Note 3, Fig. 5a. As discussed in that section, the most prominent features in the anticrossing region are: (i) the directly excited spin SW mode at the center of the crossing, and (ii) the polaron branch of the high- Q mode with lower frequency, visible slightly above -34 mT.

In addition to these main features, several very weak and narrow spectral lines are observed, which also show a magnetic field dependence similar to that of the directly excited SW mode. However, these modes do not exhibit clear coupling to the SAW and are not consistent with PSSW modes at the same in-plane wavevector as the SAW, as verified by the calculated PSSW dispersions in Fig. 2e) of the main text.

Their origin is not fully understood. Possible explanations include weak excitation of SWs with different wavevectors due to stray RF magnetic fields generated by the microstructure, or modified pinning conditions at the YIG/ZnO interface caused by surface roughness or defects introduced during ZnO sputtering.

Given their low signal amplitude, absence of clear hybridization, and lack of influence on the primary features used in our frequency-domain analysis, these modes were not included in the quantitative evaluation. As they do not contribute to the observed SAW–SW coupling, extracting individual coupling strengths from them is not meaningful within the context of this work.

To make this point clearer to the reader, we have added the following paragraph to Supplementary Note 3: “ Additional magnetic-field–dependent spectral features are observed in the anticrossing region. The upper edge of the polaron branch associated with the high- Q mode at lower frequency appears at a slightly higher field than the direct SW excitation. When the field is decreased in this region, several narrow spectral lines emerge, which shift systematically with magnetic field.

The exact origin of these features is unclear. They cannot be attributed to PSSWs with the same in-plane wavevector as the SAW, as verified by the calculated PSSW dispersions shown in Fig. 2e) of the main text. One possibility is that these signals originate from weakly excited SW modes with different wavevectors, potentially driven by different stray magnetic fields generated by the microstructure. Alternatively, they may arise from modified boundary conditions at the YIG/ZnO interface, possibly due to surface roughness or defects

introduced during ZnO sputtering, which can alter the magnetic pinning.

However, these features are low in amplitude and do not display clear hybridization behavior or anticrossing with the SAW mode. Importantly, they do not affect the frequency-domain analysis of the main anticrossing features and were therefore excluded from the quantitative evaluation presented in this work. ”

Further, we now address this paragraph and the additional modes in the main text: “ To fully characterize the coupling regime, we determined the magnon loss rate (κ_m) by analyzing the $n = 0$ SW mode, yielding $\kappa_m/2\pi = (1.24 \pm 0.02)$, MHz. A detailed description of this evaluation, together with a discussion of additional magnetic features in the anticrossing region, is provided in Supplementary Note 3 ”

3. The deviation of g between the theory and experiments for ϕ at around $\pi/2$ (Fig. 3) seems significant. How were the experimental error bars obtained in this section? Could the discrepancy between the theoretical and experimental coupling strengths be further analyzed using, e.g., finite element simulations?

The experimental error bars in Fig. 3g were obtained by fitting the observed anticrossing features to Eq. (1) of the main text, which has been revised following the suggestions of Reviewer #2. The error bars represent the standard deviations of the fitted parameters, determined through a nonlinear least-squares regression. For angles around $\phi = \pi/2$, the coupling strength g becomes very small, leading to a correspondingly reduced spectral splitting until the anticrossing is no longer observable. Including the magnon and phonon damping rates in the fitting model significantly reduced the associated uncertainties in this regime.

To clarify the origin of the error bars for the reader, we added the following sentence to the main text: “The error bars reflect the standard deviations of the fitted parameters as determined by the nonlinear least-squares regression.”

For the comparison between theory and experiment around $\phi = \pi/2$, we attempted to improve the agreement by performing finite-element simulations of the elastic strain fields (following the approach described in Phys. Rev. Applied **23**, 034062). However, this did not lead to better agreement (see Fig. R1). The FEM eigenmode corresponds to a propagating wave, whereas in our experiment we investigate a standing SAW due to the resonator. Apparently, the strain profile of propagating and standing SAWs can deviate significantly. To capture this quantitatively a full FEM modeling of the millimeter-scale resonator would be required, which is very challenging due to the large lateral dimensions of the resonator combined with the small film thicknesses and the inaccessibility of the exact material parameters which greatly influence the resulting strains. Our employed analytical theoretical approach models the experimental observations better than the FEM model. It furthermore allows us to infer the strains by comparing the experimental data with the theoretical expressions and we find the inferred strains to be in good agreement with expectations from literature.

We furthermore note that in our theoretical model the modes are assumed to be confined by the resonator (see Supplementary Note 7). As the coupling decreases, the polaron branches are no longer observable (at angles around $\phi \approx \pi/2$), and the non-hybridized magnon in the experiment is no longer limited to the resonator area (see Supplementary Note 7 for details). Consequently, the model is inaccurate in this regime. A more accurate

FIG R 1. Coupling strength g as a function of angle ϕ . Shown are the experimentally obtained values ($g_{n=0}$), the theoretical curve shown in the main text (g_{theory}), and the curve calculated using the strains from a simulated SAW eigenmode of the multilayer (g_{FEM}).

description would need to account for the evolution of the mode profiles with ϕ , reflecting changes in confinement and mode character when transitioning from strong to weak coupling.

We also believe that a more detailed theoretical framework, combining finite-element analysis with coupled magnetoelastic boundary conditions, could ultimately resolve the remaining discrepancies between theory and experiment. However, such an effort lies well beyond the scope of the present work, in which we have deliberately employed the simplest model that captures the key experimental features.

4. The study in *Commun. Phys.* 5(1), 115 (2022) reported coherent oscillations between magnons and phonons, while that in *Phys. Rev. Lett.* 130, 193603 (2023) reported swap oscillations between magnons and superconducting qubits. Did the Rabi-like oscillations observed in this work differ from the previous results? Furthermore, can consistent information—such as coupling strengths and magnon decay rates—be extracted in the Rabi-like oscillations signal?

In contrast to the Rabi oscillations reported in the referenced works, the oscillations observed here arise during the transient response of the resonator, which makes the dynamics more intricate. In *Commun. Phys.* 5(1), 115 (2022), a 100 fs laser pulse excites the magnons, after which coherent oscillations between quasiparticles occur. In our system, the coupling takes place inside a SAW cavity. The SAW pulse — visible in the time domain (obtained via inverse Fourier transformation from quasi-CW frequency-domain data) — is reflected by the Bragg mirrors and passes the resonator on time scales comparable to the Rabi period. In *Phys. Rev. Lett.* 130, 193603 (2023), the reported Rabi-like oscillations occur between a transmon qubit and the ferromagnetic resonance (FMR) magnon in a YIG film. There, the $k = 0$ magnon couples to an electromagnetic field propagating at the speed of light, and the probability amplitudes of the quantum states can be accessed directly via the qubit. Due to the light speed group velocity of the electromagnetic wave, the resonator reaches its steady state almost immediately. In contrast, in our SAW–SW system, the comparable group velocities of magnons and phonons inherently produce a resonator transient response. This scenario is unique to our system and is not captured by the simple two-level Rabi model; a rigorous theoretical description would require a full time-dependent modeling of the coupled transient SAW resonator response, which is beyond the scope of this work.

It is indeed possible to extract the coupling strength and decay rates from the time-domain data; however, certain caveats arise due to the signal quality and the use of a simplified theoretical model. To address this, we have added an additional plot and corresponding analysis in Supplementary Note 4, demonstrating that both the coupling strength and the spin-wave decay rate can be obtained from the time-domain data. The resulting values are in good agreement with expectations and with the frequency-domain analysis presented in the main text.

5. The coupling between phonons and magnon polaritons in cavity magnomechanics systems has extensively been studied in, e.g. PRX 11, 031053 (2021) and Nat. Commun. 16, 5652 (2025). Notably, the experiment in the Nat. Commun. 16, 5652 (2025) achieved strong coupling between magnon polaritons and phonons by engineering polariton dissipation, despite the intrinsically weak magnon-phonon coupling strength. Could this dissipation-engineering approach be combined with this work to further enhance the cooperativity between SAWs and SWs?

We thank the reviewer for bringing up these interesting ideas.

In general, magnon-polaritons correspond to zero- k vector ferromagnetic resonance (FMR) magnons coupled to an electromagnetic cavity. This is also the case for the experimental system reported in Nat. Commun. 16, 5652 (2025), where a two-port electromagnetic cavity was employed to directly tune the external cavity dissipation rates (denoted κ_1 and κ_2) alongside a separate port for direct FMR excitation. When the condition $\kappa_1 + \kappa_2 - \kappa_{\text{int}} = \kappa_{\text{m}}$ is fulfilled—where κ_{int} is the internal cavity loss rate and κ_{m} is the magnon loss rate—the polariton decay rate is given by $\kappa_{\pm} = \frac{-\kappa_1 - \kappa_2 + \kappa_{\text{int}} + \kappa_{\text{m}}}{2} = 0$. This condition, referred to as coherent perfect absorption (CPA), has been realized in macroscopic phononic systems (Phys. Rev. Lett. 135, 06700 (2025), J. Acoust. Soc. Am. 140, EL94–EL100 (2016)) but (to our knowledge) has not yet been demonstrated in SAW devices.

To achieve CPA in our system, one would require two-port SAW resonators whose external Q factors for both ports are reduced to satisfy the above relation. For illustration (using realistic values close to those in the main text): assuming $\kappa_{\text{m}} = \kappa_{\text{int}} = 1$ MHz, we would require $\kappa_1 = \kappa_2 = 1.5$ MHz. For a base frequency of 1.5 GHz, this corresponds to an external quality factor $Q_{\text{ext}} = 1000$ for each SAW port. In our current devices, Q_{ext} is significantly larger—ranging from 3000 to 12000 depending on IDT design. While a Q_{ext} of 1000 is achievable by for example increasing the finger number of the IDT, reaching it would require careful redesign of the resonator IDTs to precisely adjust Q_{ext} , which would also affect Q_{int} .

We hope this answers the question.

We have added the references (PRX 11, 031053 (2021) and Nat. Commun. 16, 5652 (2025)) to the main text and wrote: “The low dissipation rate of the bosonic quasiparticles in ordered magnets, namely magnons, has facilitated the formation of their hybrids with photons (magnon polaritons) [12–18] and with phonons (magnon polarons) [19–24]. Even quasiparticles combining all three fundamental excitations have been demonstrated in po-

laromechanical systems [25]. ”

Further, we modified the discussion and conclusion section: “ Moreover, by implementing dissipation engineering strategies [18, 25], it may become possible to further suppress magnon–polaron losses and extend the coupling to additional excitations in solid-state platforms, thereby broadening the potential of this on-chip approach. ”

6. Could you please provide further clarification on how the number of gated passes affects resonator loss? It includes how to experimentally control the number of gated passes.

As the number of included passes in the time-domain gating increases, the bandwidth of the resonator signal decreases. This is reflected in an increase of the Q factor, which serves as a figure of merit describing the ratio between the resonance frequency and the bandwidth. A higher Q factor corresponds to a lower effective loss rate in the resonator. Thus, by including more SAW passes in the time window, one effectively isolates the longer-lived, lower-loss components of the resonator response.

In our work, time gating is applied as a post-processing technique. The number of gated passes can be adjusted by selecting a specific time window in the time-domain signal before performing a Fourier transform back into the frequency domain.

However, this behavior can also be accessed directly in time-resolved experiments. For example, using a microwave source in combination with a fast microwave switch and a real-time oscilloscope, the transient evolution of the SAW resonator response could be directly measured. Optical techniques such as timeresolved microfocused Brillouin light scattering (BLS) provide another route. In this approach, a microwave pulse of defined length and frequency is applied to the resonator structure, and the BLS measurement window is positioned at the end of the pulse. By sweeping the excitation frequency and varying the pulse duration, one can experimentally track the temporal evolution of the resonance linewidth and hence the effective loss rate as a function of pass number.

To address this question comprehensively, we have added Supplementary Note 9, which includes additional data analysis and a detailed discussion of how the number of gated passes influences the effective resonator loss and how this can be experimentally controlled.

Reviewer #2 (Remarks to the Author):

This work presents the hybridization of magnons in single-crystalline yttrium iron garnet (YIG) films with phonons confined in a zinc oxide (ZnO)-based surface acoustic wave (SAW) resonator. The study demonstrates tunable coupling strength via magnetic field direction, achieving strong magnon-phonon coupling through high-Q SAW modes in the GHz regime with low damping rates (few MHz). A phenomenological model is introduced to describe this interaction, incorporating spatial mode profiles. Rabi-like oscillations are observed in the time domain, providing direct evidence of strong magnon-phonon interaction shaping the temporal response of the SAW resonator. This work is establishing a promising platform for hybrid spin-acoustic systems in extended magnetic materials. However, several areas require clarification and improvement:

We thank the reviewer for their positive assessment of our work as well as the constructive feedback. We respond to the comments in a point-wise manner below and have revised the manuscript accordingly.

1. Confusing Link Between Damping Mechanisms (Line 060): The text suggests that the limitation of SAW devices stems from large phonon damping, yet proposes low magnon damping in YIG as a solution. This logical jump is unclear—low magnon damping does not inherently mitigate phonon losses. Please clarify how YIG’s low damping addresses the phonon-related limitations.

The sentence in question was indeed poorly worded. In the cited work ([21]), the relaxation rate limiting the device performance was that of the spin-wave (SW) system, not the SAW. The SW-hosting material used in that study was CoFeB, which exhibits relatively high damping in comparison to YIG.

To remove the ambiguity, we revised the sentence to: “ Recent reports using surface acoustic wave (SAW) devices are promising, but remain limited by high spin wave (SW) relaxation rates ($\kappa_m/2\pi \approx 50$ MHz [22]), which preclude time-domain studies and other coherent control schemes. ”

2a. Unclear Concept of Magnon Confinement (Fig. 1 and Caption): The claim that magnons are not confined in the weak coupling regime is problematic. Since magnons are excitations within the YIG film, their spatial confinement should not depend on the coupling strength. What does it mean physically for the magnon wavefunction to “escape” the cavity? Please provide experimental or theoretical justification for this assertion.

Our device consists of a magnet/piezo bilayer that is laterally very large (practically infinite) in the planar directions. Metallic electrodes are deposited on top of the 980 nm-thick ZnO piezoelectric layer, imposing boundary conditions for the elastic strain fields but not for the magnetization. The electrodes are not in direct contact with the YIG but separated by the ZnO layer. These electrodes form an acoustic cavity: elastic waves that fulfill the Bragg condition are reflected, while those that do not can propagate out of the resonator, as the material extends beyond the resonator boundaries. In contrast, pure magnons (magnons that are not coupled to a phonon mode) are unaffected by the electrodes, which do not impose meaningful boundary conditions on the magnetization. As a result, all pure magnons can freely traverse the Bragg mirrors of the acoustic resonator and are not reflected. In contrast to phonons, they are not laterally confined to the resonator.

This situation is fundamentally different from the case mentioned by the reviewer of a magnon-photon system [IEEE Trans. Magn. 52, 1000107 (2016)], where a magnetic sample is placed in a 3D microwave cavity. There, the magnon is naturally confined to the magnetic sample, while the photon is confined by the cavity geometry, so excitation leakage does not play a significant role. In contrast, in our system, the patterned electrodes on the piezo/magnet bilayer define an acoustic cavity much smaller than the overall sample area.

To clarify this distinction, we have revised Fig. 2 and added additional clarification to the main text: “ Our acoustic resonator is defined on an extended bilayer sample (Fig. 2b)) and thus occupies a small fraction of the available space to which the hybridized excitations are confined by the resonator. This contrasts sharply with the typical concept and design in magnon-photon systems [39,40], in which the magnons and photons are confined separately by their respective full sample spaces. ”

2b. Ambiguity in “Hybridization-Driven Confinement” (Line 89 & Ref. 31): The idea that hybridization is necessary for magnon confinement is not clearly substantiated. Drawing a parallel with exciton-polaritons (Ref. 31) is insufficient without rigorous support. Please explain the physical mechanism by which coupling enhances confinement and specify the physical parameter that defines such a confinement.

We thank the reviewer for pointing out the lack of justification for our comparison to the exciton-polariton case. In our theoretical approach, we treat both magnons and phonons as confined within the volume of the SAW resonator. This assumption leads to good agreement with the experimentally observed coupling strength in the strong coupling regime. In contrast, in the weak coupling regime, the theory predicts a larger coupling strength than observed experimentally. We interpret this discrepancy as follows: in the strong coupling regime, the phonon and magnon cease to exist as independent excitations and instead form a magnon-polaron. The experimentally observed coupling strength of this hybrid mode is in good quantitative agreement with our theory, which assumes confinement of the interacting modes to the resonator. In the weak coupling regime, however, only the phonons are confined by the resonator. The weak coupling of magnons to the phonon mode in this case is too weak to yield meaningful confinement of magnons to the phononic resonator. Thus, the elastic boundary conditions of the acoustic cavity confine the hybrid magnon-polaron modes in the strong coupling limit but do not confine magnons in the weak coupling regime. We note that for our low-damping YIG, the propagation length of magnons at the relevant wavelength ($500\ \mu\text{m}$, see Supplementary Note 4) is comparable to the resonator dimensions. To avoid ambiguity, we have removed the following sentences from the main text: “Consequently, a magnon-polaron cavity is realized, in which the quasiparticle hybridization is a necessity for confinement. Our realized approach is reminiscent of the recently observed self-hybridized exciton polaritons [31], where the exciton-photon coupling plays a similar important role in photon confinement.”

and

“Furthermore, this confinement and the consequent formation of magnon-polarons are similar in spirit to the formation of self-hybridized exciton polaritons [31] due to the efficient photonic confinement via its coupling to the excitonic mode.”

We now provide a more detailed discussion and additional analysis of this point in Supplementary Note 7.

2c. Ambiguous Discussion of Confinement (Line 212): The statement that hybridization induces confinement is vague. Since magnons remain spatially within YIG, it is unclear what additional “confinement” is being claimed. Moreover, later discussions (e.g., Line 299) suggest the variation in coupling strength arises from spin wave ellipticity, not spatial effects. The link between ellipticity and confinement needs clarification. I would like to mention that angular-dependent coupling strength was previously used for controlling the magnon-photon coupling, and it governs by the polarization of the photon mode instead of mode confinement [see, IEEE Transactions on Magnetics, 52, 1000107 (2016).]

Regarding the point of additional confinement of magnons beyond the physical dimensions of the YIG layer, please refer to our reply above: While magnons are always confined to the YIG layer, only the hybridized magnon–polaron modes are additionally confined by the acoustic resonator.

In our theoretical model, we assume that both magnons and phonons are confined to the resonator area. This allows us to capture the reduction of coupling strength with increasing angle ϕ due to the change in spin-wave ellipticity (or more general the mode profile). However, the model does not accurately reproduce the full extent of the experimentally observed decrease of the coupling strength near $\phi = \pi/2$. We attribute this discrepancy to the fact that, in the weak-coupling regime where magnons exist in a nonhybridized form, they are not confined by the resonator, reducing the resulting coupling further.

To make this point clearer in the manuscript, we have revised the relevant paragraphs as follows:

Line 212 (main text): “Our observed coupling strength agrees well with the calculations of our theoretical model, which assumes that the magnon mode is restricted to the magnetic volume within the resonator. This indicates that, in the strong coupling regime, the resonator confines the hybridized magnon–polaron and thereby gives rise to new coupled magnetoelastic boundary conditions [40,41].”

Line 299 (main text): “As ϕ increases, the experimentally observed coupling strength [Fig. 3g)] decreases more strongly than predicted by our theoretical model, which assumes the phonon and magnon to be confined to the area of the resonator. This additional reduction of coupling strength is in accordance with an increased uncoupled magnon mode volume due to magnon leakage out of the acoustic resonator as depicted schematically in Fig. 1 and discussed in more detail in Supplementary Note 7.”

We have also cited the reference suggested by the reviewer in this context and added

additional discussion to this topic in Supplementary Note 7.

Regarding the angular dependency of coupling strength due to the change of polarization in the spirit of [IEEE Transactions on Magnetics, 52, 1000107 (2016).], our model takes the corresponding effect, namely the angular dependency of the magneto-acoustic interaction, already fully into account as discussed in Supplementary Note 7C.

Indeed, this angular dependency is more complex than the case of magnon-photon coupling (where, in the linear regime, coupling is only possible if the oscillating magnetic field is orthogonal to the equilibrium magnetization). The angle between magnetization direction and SAW direction forms a characteristic fingerprint of magnetoacoustics (see e.g. Phys. Rev. Applied 15, 034046 (2021)). Considering this angular dependency is absolutely required to match the model to our data.

But importantly, it is not sufficient to explain the angular dependency of the observed coupling strength on its own. In our model, we must additionally consider the change of ellipticity. Indeed, in the strongly coupled case, the combination of magnetoacoustic symmetry and ellipticity then provides quantitative agreement with the experiment when using reasonable parameters (t_x, t_y, t_z, d) that only describe the acoustic mode profile (see Supplementary Note 7, 7E). All other parameters are either taken from literature, directly measured, or obtained from micromagnetic simulations. Notably, the model does not rely on an overall scaling factor.

As discussed above, in the weakly coupled regime our model still overestimates the coupling strength compared to the experimental result. This is in agreement with reduced mode confinement for increased ϕ .

We note that it would of course be desirable to model the implied change of confinement with ϕ quantitatively. However, we cannot measure the extent of the magnon confinement (effective magnon delocalization area A_m compared to the known acoustic resonator area A) experimentally because we could not spatially resolve the magnons, presumably due to the very weak magnon BLS signal in the ZnO/YIG bilayer. This precludes a meaningful introduction of a phenomenological angle-dependent factor $\sqrt{A/A_m}$ into the model (see Supplementary Note 7D).

3. Model Limitations and Suggested Revision (Eq. 1 and Fig. 3): Equation 1 models the strong coupling regime but fails in the weak coupling region (e.g., near 90° field angle in Fig. 3). To more accurately fit the data, the model should include damping effects, as shown in Eq. (2) of J. Appl. Phys. 129, 201101 (2021). Revising Eq. 1 accordingly would strengthen the analysis.

We thank the reviewer for this helpful suggestion. We have revised Eq. (1) and Figs. 2 and 3 accordingly in the updated manuscript, citing the recommended reference. This refinement significantly reduces the fitting error, particularly in the regime of low coupling strength, while leaving the extracted coupling strength unchanged within the fit error. Thus, the revision also leaves our corresponding scientific conclusions unchanged. The small difference in extracted coupling strengths between the previous and revised fitting equations is illustrated in Fig. R2.

FIG R 2. The upper panel displays the fitted coupling strengths obtained with both fit models, where the revised fit incorporates the magnon and phonon relaxation rates shown in the main text. The lower panel shows the difference between the two approaches, highlighting that the deviation in coupling strength is minimal, which can be partly attributed to the similar decay rates of magnons and phonons.